# IL-4 Signaling Promotes Myoblast Differentiation and Fusion by Enhancing the Expression of MyoD, Myogenin, and Myomerger

**DOI:** 10.3390/cells12091284

**Published:** 2023-04-29

**Authors:** Mitsutoshi Kurosaka, Yung-Li Hung, Shuichi Machida, Kazuhisa Kohda

**Affiliations:** 1Department of Physiology, St. Marianna University School of Medicine, Kanagawa 216-8511, Japan; kkohda@marianna-u.ac.jp; 2Institute of Health and Sports & Medicine, Juntendo University, Chiba 270-1695, Japan; y-hon@juntendo.ac.jp (Y.-L.H.); machidas@juntendo.ac.jp (S.M.); 3Graduate School of Health and Sports Science, Juntendo University, Chiba 270-1695, Japan

**Keywords:** interleukin-4, myoblast fusion, myogenic regulatory factors, muscle differentiation, myomaker, myomerger, NFATc2

## Abstract

Myoblast fusion is essential for skeletal muscle development, growth, and regeneration. However, the molecular mechanisms underlying myoblast fusion and differentiation are not fully understood. Previously, we reported that interleukin-4 (IL-4) promotes myoblast fusion; therefore, we hypothesized that IL-4 signaling might regulate the expression of the molecules involved in myoblast fusion. In this study, we showed that in addition to fusion, IL-4 promoted the differentiation of C2C12 myoblast cells by inducing myoblast determination protein 1 (MyoD) and myogenin, both of which regulate the expression of myomerger and myomaker, the membrane proteins essential for myoblast fusion. Unexpectedly, IL-4 treatment increased the expression of myomerger, but not myomaker, in C2C12 cells. Knockdown of IL-4 receptor alpha (IL-4Rα) in C2C12 cells by small interfering RNA impaired myoblast fusion and differentiation. We also demonstrated a reduction in the expression of MyoD, myogenin, and myomerger by knockdown of IL-4Rα in C2C12 cells, while the expression level of myomaker remained unchanged. Finally, cell mixing assays and the restoration of myomerger expression partially rescued the impaired fusion in the IL-4Rα-knockdown C2C12 cells. Collectively, these results suggest that the IL-4/IL-4Rα axis promotes myoblast fusion and differentiation via the induction of myogenic regulatory factors, MyoD and myogenin, and myomerger.

## 1. Introduction

Skeletal muscle comprises approximately 40% of the human body mass and is required for essential functions, such as locomotion and metabolism [1]. Loss of skeletal muscle is often associated with pathological conditions, such as muscle atrophy and metabolic diseases [2]. Thus, skeletal muscle maintenance has been recognized as a determinant of the quality of life [3]. Skeletal muscle is composed of large multinucleated cells containing many post-mitotic myonuclei. During muscle formation and regeneration, myoblasts fuse with each other and with multinucleated myotubes to form myofibers [4,5,6]. Inhibition of myoblast fusion impairs muscle formation and regeneration [7,8,9,10]. Therefore, it is important to elucidate the mechanisms underlying myoblast fusion to maintain skeletal muscle mass and its function and to understand the pathophysiology of muscle disorders.

Recently, two membrane proteins—myomaker and myomerger (also known as myomixer or minion)—that drive myoblast fusion have been identified [11,12,13], and intensive studies have revealed their essential roles in myogenesis and muscle regeneration [7,9,11,12,13,14]. At present, a two-step cell fusion model has been proposed. First, myomaker fuses the contacting monolayers of the two myoblast membranes (hemi-fusion), and then, myomerger drives pore formation and expansion [15,16]. These two muscle fusion proteins have been implicated in the pathogenesis of muscle disorders. For example, the reduced expression of myomaker due to autosomal recessive mutations in the myomaker gene causes a congenital myopathy, Carey–Fineman–Ziter syndrome [17] and reduced expression of myomaker and myomerger proteins cause muscular pathology in spinal muscular atrophy [18]. During myogenesis, myomaker and myomerger share similar expression patterns with myoblast determination protein 1 (MyoD) and myogenin, members of the myogenic regulatory factors [11,14], and these proteins are known to regulate the expression of myomaker and myomerger [19,20,21]. However, it is still unclear what kind of extracellular signals control their expression and function.

IL-4 is a cytokine that, when bound to the IL-4 receptor (IL-4R), activates or inactivates transcription factors through its tyrosine kinase activity, thereby causing various physiological responses in different cellular systems, including immune cells, epithelial cells, and fibroblasts [22]. In skeletal muscle cells, a transcription factor, nuclear factor of activated T cells (NFAT) c2, stimulates the production of IL-4, which mediates myoblast fusion [23,24]. We have previously shown that IL-4 treatment increases the number of myonuclei per myotube in primary-cultured myoblast cells [25,26], thereby indicating enhanced myoblast fusion. Although this finding suggests that IL-4/IL-4R signaling regulates myoblast fusion, the underlying mechanisms remain unclear. This study aims to investigate whether IL-4 is involved in myoblast fusion and differentiation by regulating the expression of the transcription factors—MyoD and myogenin—and their targets, myomaker and myomerger. 

## 2. Materials and Methods

### 2.1. Cell Culture 

C2C12 myoblast cells (ATCC, Manassas, VA, USA) were maintained in growth medium (GM) consisting of Dulbecco’s modified Eagle’s medium (DMEM; Life Technologies Inc., Carlsbad, CA, USA) supplemented with 10% bovine growth serum (Cytiva, Tokyo, Japan) and 1% penicillin–streptomycin (Life Technologies Inc.). To induce the differentiation of myoblast cells, GM was replaced with differentiation medium (DM), DMEM supplemented with 2% horse serum (Life Technologies Inc.) and 1% penicillin–streptomycin.

Recombinant IL-4 (R&D Systems, Minneapolis, MN, USA) was used as previously described [25,26]. IL-4 was dissolved in distilled water containing 0.1% bovine serum albumin (BSA; Wako, Tokyo, Japan) and added to the culture medium at a final concentration of 1 to 100 ng/mL.

### 2.2. 5-Ethynyl-20-Deoxyuridine (EdU) Incorporation Assay 

C2C12 cells were grown in GM for 24 h. For the last 2 h, cells were labeled with EdU and stained using the Click-iT Plus EdU cell proliferation assay kit (Alexa Fluor 488; Life Technologies Inc.) [27]. Cell nuclei were counterstained with Hoechst 33342 (Life Technologies Inc.) for 30 min at room temperature. Images of EdU incorporation were visualized using a fluorescence microscope (BZ-9000; Keyence Co., Osaka, Japan), and the number of EdU-positive cells was counted in four randomly selected microscopic fields per sample. The average values obtained from each sample were used for analysis.

### 2.3. Transfection of Small Interfering RNAs (siRNAs) and Expression Vectors 

IL-4Rα-specific Dicer substrate siRNA (siIL-4Rα), NFATc2-specific Dicer substrate siRNA (siNFATc2), and control siRNA (siCtrl) were purchased from Integrated DNA Technologies (IDT; San Diego, CA, USA) [28]. SiRNAs were transfected using Lipofectamine RNAi MAX reagent (Life Technologies Inc.) according to the manufacturer’s instructions. After transfection, C2C12 cells were maintained in GM for 24 h. The medium was then changed to fresh GM or DM to induce myoblast proliferation or fusion for subsequent experiments.

For the expression of myomerger in the IL-4Rα knockdown C2C12 cells, we first transfected siIL-4Rα or siCtrl into the cells. One day later, myomerger expression vector or empty vector (VectorBuilder, Chicago, IL, USA) was transfected using Lipofectamine 3000 (Thermo Fisher Scientific, Waltham, MA, USA) according to the manufacturer’s instructions. The cells were maintained in GM for another 24 h. Then, the culture medium was replaced with fresh DM. 

### 2.4. Quantitative Real-Time PCR (qRT-PCR) 

The details of the protocol have been described previously [26]. Briefly, total RNA was isolated from C2C12 cells using an RNA extraction reagent (Sepasol-RNA I Super G; Nacalai Tesque Inc., Kyoto, Japan) and RNA mini-columns (Favorgen Biotech Corp., Ping-Tung, Taiwan), according to the manufacturer’s protocol. First-strand cDNA for PCR was generated using ReverTra Ace qPCR Master Mix (Toyobo Co., Ltd., Osaka, Japan). Quantification of mRNA expression was performed using a real-time PCR system (Step One Plus; Life Technologies Inc.) with Syber Green Master Mix Reagent (Toyobo Co., Ltd.). For delta–delta Ct analysis, glyceraldehyde-3-phosphate dehydrogenase (GAPDH) mRNA was used as an internal reference. The primers used in this study were purchased from IDT, and their sequences are shown in the Appendix A.

### 2.5. Immunocytochemistry 

SiRNA-transfected C2C12 cells were subjected to immunocytochemical analysis as previously described [26]. Briefly, C2C12 cells were washed with phosphate-buffered saline (PBS) and fixed in ice-cold absolute methanol for 15 min. After several washes with PBS, cells were incubated in blocking solution (PBS supplemented with 3% BSA) for 30 min and treated with primary antibodies against myosin heavy chain (MyHC; 1:50; Development Studies Hybridoma Bank (DSHB), Iowa City, IA, USA), MyoD (1:200; Santa Cruz Biotechnology Inc., Dallas, TX, USA), and myogenin (1:50; DSHB) in blocking solution overnight at 4 °C. After washing with PBS, C2C12 cells were probed with the appropriate secondary antibodies conjugated to Alexa Fluor 488 or 546 (Life Technologies Inc.) in blocking solution for 1 h at room temperature. Nuclear counterstaining was then performed using 4,6-diamidino-2 phenylindole (DAPI; Sigma-Aldrich, St. Louis, MO, USA). Images of stained C2C12 cells were captured using a microscope (BZ-9000; Keyence Co., Osaka, Japan) and analyzed using Fiji software [26]. The number of myonuclei per myotube and myosin-positive cells were counted. The differentiation index was calculated as the number of nuclei in MyHC-positive cells divided by the total number of nuclei. These parameters were obtained from 4 randomly selected microscopic fields per sample and their average values were used for further analysis.

### 2.6. Western Blotting Analyses 

Cultured C2C12 cells were homogenized in ice-cold radioimmunoprecipitation assay (RIPA) buffer (Wako) with protease and phosphatase inhibitors (Nacalai Tesque Inc.) and processed for western blotting analyses as previously described [25,26]. Primary antibodies used were against phosphorylated MyoD (1:1000; Santa Cruz Biotechnology Inc.), myogenin (1:1000; DSHB), myomaker (1:1000, Abcam, Cambridge, UK), myomerger (1:1000; R&D Systems), IL-4Rα (1:1000; Santa Cruz Biotechnology Inc.), MyHC (1:200; DSHB), and GAPDH (1:2000; Cell Signaling Technology Inc., Danvers, MA, USA). Luminescence signals from ECL reagent (Bio-Rad Laboratories Inc., Hercules, CA, USA) were detected using LAS-4000 (Fujifilm Corp., Tokyo, Japan). Quantitative densitometric analyses were performed using Fiji software.

### 2.7. Cell Mixing Assay 

To examine myoblast–myoblast fusion, the siRNA-transfected C2C12 cells were labeled with fluorescent lipid Dil-Red or Green CMFDA cell tracker (Thermo Fisher Scientific). Equal numbers of cells labeled with each color were co-cultured in DM for 72 h and fixed. The number of double-labeled cells was counted in four randomly selected microscopic fields per sample, and the average values were used for further analysis.

### 2.8. Statistical Analyses 

Data are presented as mean ± standard deviation (SD). One-way analysis of variance (ANOVA) followed by the Tukey–Kramer multiple comparison test was used for the evaluation of multiple group data. For comparisons between the two groups, unpaired Student’s *t*-tests were used. All analyses were performed with Prism v.9.0 (GraphPad Software Inc., San Diego, CA, USA). Statistical significance was set at *p* < 0.05.

## 3. Results

### 3.1. IL-4 Promoted Myoblast Fusion and Differentiation, and Increased the mRNA Expression of Myogenic Regulatory Factors and Myomerger in C2C12 Cells

We first confirmed that IL-4 promotes myoblast fusion. C2C12 myoblast cells were treated with 1, 10, or 100 ng/mL IL-4 in DM—a low-serum differentiating condition—for 72 h, and immunocytochemical analysis was performed to examine myotube formation. Consistent with our previous studies [26], the number of myonuclei per myotube significantly increased by incubation with 10 ng/mL IL-4, thereby indicating that myoblast fusion was promoted (Figure 1A,B). Therefore, we used 10 ng/mL IL-4 in the following experiments. 

We investigated whether IL-4 promoted myoblast differentiation by evaluating the differentiation index (see Materials and Methods). The differentiation index of IL-4-treated C2C12 cells was significantly higher than that of untreated cells after 72 h of incubation in DM (Figure 1C). Since myoblast differentiation inhibits their proliferation [29], we performed EdU incorporation studies to know the effects of IL-4 on myoblast proliferation. The percentage of EdU-positive C2C12 cells significantly decreased by IL-4 treatment when compared to that of control cells after 24 h incubation in GM, a full-serum proliferating condition (Figure 1D,E). These results suggest that IL-4 inhibits cell cycle progression and promotes differentiation of C2C12 myoblast cells.

Next, we examined the effects of IL-4 on the expression of molecules essential for myoblast differentiation and fusion, such as MyoD, myogenin, myomaker, and myomerger. After treatment with IL-4 in DM for 72 h, C2C12 myoblasts were harvested to analyze the mRNA levels of these molecules by qRT-PCR. The results showed that IL-4 treatment significantly increased the mRNA expression of MyoD, myogenin, and myomerger (Figure 1F). However, no significant difference was found in the mRNA level of myomaker. These findings suggest that IL-4 signaling should regulate the expression of myogenic regulatory factors and myomerger, which should promote myoblast fusion and differentiation. 

### 3.2. IL-4Rα Knockdown Impaired Fusion and Differentiation of C2C12 Cells and Reduced the Expression of Myogenic Regulatory Factors and Myomerger

IL-4 exerts its effects by binding to an IL-4Rα subunit of IL-4R and activating its signaling cascade. Therefore, a loss of IL-4Rα should diminish the effects of IL-4 treatment. We knocked down IL-4Rα expressed in C2C12 myoblast cells by transfecting siRNAs (Appendix A) and analyzed the number of myonuclei per myotube and the differentiation index. The number of nuclei per myotube was significantly reduced in the IL-4Rα knockdown (KD) group when compared to the control (Ctrl) group (transfected with control siRNA) and the IL-4 group (transfected with control siRNA and treated with IL-4; Figure 2A,B). Treatment of IL-4RαKD C2C12 cells with IL-4 did not restore the reduced number of nuclei per myotube (IL-4RαKD + IL-4 in Figure 2A,B). In addition, when we compared the number of nuclei in the MyHC-positive cells, the IL-4RαKD and IL-4RαKD + IL-4 groups showed a significant increase in the percentage of MyHC-positive cells with a single nucleus and a significant decrease in the percentage of the cells with three or more nuclei (Figure 2C). The differentiation indices and the expression levels of MyHC were significantly lower in the IL-4RαKD and IL-4RαKD + IL-4 groups than in the Ctrl and the IL-4 groups (Figure 2D–G). Importantly, all of these parameters described above were not significantly different between the IL-4RαKD and IL-4αKD + IL-4 groups. These results suggest that the effects of IL-4 on myoblast fusion and differentiation were mediated by IL-4Rα. A previous study showed that IL-4 was secreted from myotubes [24]. Since myotubes were well present in the Ctrl culture dish when compared to those in the IL-4RαKD and IL-4RαKD + IL-4 groups (Figure 2A), it is likely that endogenously secreted IL-4 exerted its effects on the myoblasts in the Ctrl group. This could be the reason why the IL-4RαKD and IL-4αKD + IL-4 groups showed significantly lower levels of myoblast fusion and differentiation than the Ctrl group. 

Given that IL-4RαKD impairs differentiation and fusion of C2C12 myoblast cells, IL-4RαKD should suppress the enhanced expression of MyoD, myogenin, and myomerger due to the reduction of IL-4 signaling. Indeed, immunocytochemical analysis showed that the number of myogenin-positive C2C12 cells in the IL-4RαKD and IL-4RKD + IL-4 groups was significantly lower than that in the IL-4 and Ctrl groups (Figure 3A,B). Furthermore, through qRT-PCR studies and western blotting analyses, we found significantly reduced expression levels of MyoD and myogenin in the IL-4RαKD and IL-4RKD + IL-4 groups when compared to those in the IL-4 group (Figure 3C–E). The expression of myomerger mRNA and protein was also significantly suppressed in the IL-4RαKD group, whereas there was no significant difference in the expression of myomaker between all the experimental groups (Figure 3C–E). These findings are consistent with the effects of IL-4 treatment in C2C12 myoblast cells, suggesting that stimulation of the IL-4/IL-4Rα axis leads to myoblast fusion and differentiation by the enhancement of the expression of myogenic regulatory factors and myomerger. 

### 3.3. Activation of IL-4 Signaling Promoted Myoblast Differentiation Even in a Cell-Proliferating Condition

So far, we have studied IL-4 signaling using C2C12 myoblast cells cultured in low-serum DM, which arrests cell growth and promotes differentiation into multinucleated myotubes. We investigated whether IL-4 signaling could also exert its differentiating effects in full-serum GM, a cell-proliferating condition. We knocked down IL-4Rα in C2C12 cells and incubated them in GM with or without IL-4 for 24 h. The number of myogenin-positive cells was significantly higher in the IL-4 group when compared to the IL-4RαKD and IL-4RαKD + IL-4 groups (Figure 4A,B). In addition, qRT-PCR studies indicated that the expression of MyoD and myogenin significantly increased in the IL-4 group (Figure 4C). The IL-4 group also showed increased myomerger mRNA expression, whereas there was no significant difference in the expression of myomaker among the experimental groups. These results demonstrate that the activation of the IL-4/IL-4Rα axis facilitates the transition to myogenic differentiation even in GM through enhanced induction of MyoD, myogenin, and myomerger. 

### 3.4. Reduced Myomerger Expression Partially Rescued Impaired Myoblast Fusion in IL-4RαKD C2C12 Cells 

The IL-4RαKD C2C12 cells showed impairment in fusion and differentiation (Figure 2). In these cells, the expression of myogenic regulatory factors, MyoD and myogenin, and myomerger significantly decreased, while the expression of myomaker was not affected (Figure 3). Although myomaker and myomerger play essential roles in the process of myoblast fusion [11,12,13], myomaker must be expressed in both the fusing cells, whereas myoblast fusion proceeds normally when myomerger is expressed on only one side of the cells [16]. Thus, reduced expression of myomerger should be a cause of impaired myoblast fusion in the IL-4RαKD C2C12 cells. To test this hypothesis, we performed a cell mixing assay. C2C12 cells were transfected with control siRNA (Ctrl) or siRNA for IL-4Rα (IL-4Rα) and labeled with red or green fluorescent lipid. Equal numbers of cells of each color were mixed and maintained in DM for 72 h (Figure 5A). The number of double-labeled cells significantly increased in the co-culture of Ctrl and IL-4RαKD C2C12 cells (Ctrl/IL-4RαKD) when compared to that of IL-4RαKD C2C12 cells (IL-4RαKD/IL-4RαKD; Figure 5A,B), whereas it did not reach the level of Ctrl/Ctrl co-cultured cells. We obtained similar results when we compared the number of nuclei per syncytia among the three groups (Figure 5C). 

To further analyze a causal role of myomerger, we performed a rescue experiment. By transfecting myomerger into the IL-4RαKD C2C12 cells, the expression of myomerger was restored, while the expression of myomaker did not significantly change (Appendix A). The number of myonuclei per myotube in the IL-4RαKD C2C12 cells with myomerger transfection (IL-4RαKD/pMyomerger) was significantly restored when compared to that in the IL-4RαKD C2C12 cells (IL-4RαKD/pEmpty). However, it was not fully recovered to the level of the control C2C12 cells (Ctrl/pEmpty; Figure 5D,E). These results suggest that a lack of myomerger in IL-4RαKD C2C12 cells should be a cause of the impaired myoblast fusion, but other factors should also be involved. 

### 3.5. Knockdown of NFATc2 Abolished the Effects of Activation of the IL-4/IL-4Rα Signaling Pathway in C2C12 Cells

Horsley et al. showed that NFATc2 stimulates the production of IL-4, which mediates myoblast fusion [24]. It has also been reported that MyoD and NFATc2 or NFATc3 cooperatively activate myogenin expression in C2C12 cells [30]. Therefore, we investigated whether NFATc2 might be involved in IL-4/IL-4Rα signaling. We first investigated whether IL-4 treatment increases NFATc2 mRNA expression in IL-4RαKD C2C12 cells. The expression of NFATc2 mRNA was significantly higher in the Ctrl and IL-4 groups than in the IL-4RαKD and IL-4RαKD/IL-4 groups (Figure 6A), indicating that IL-4Rα activation promoted NFATc2 expression. Knockdown of NFATc2 suppressed the enhanced expression of MyoD, myogenin, and myomerger mRNA by IL-4 treatment, whereas myomaker expression was unaffected (Figure 6B and Appendix A). Immunohistochemical analyses showed that the number of myonuclei per myotube and differentiation indices were significantly lower in the NFATc2KD and NFATc2KD + IL-4 groups compared to the Ctrl and the IL-4 groups (Figure 6C–E). QRT-PCR studies indicated that the expression of MyHC was suppressed in the NFATc2KD and NFATc2KD + IL-4 groups (Figure 6F). These results suggest that NFATc2, whose expression was increased by IL-4Rα activation, should regulate myoblast fusion and differentiation by enhancing the expression of myogenic regulatory factors and myomerger.

## 4. Discussion

In this study, we investigated the role of IL-4 signaling in myoblast fusion and differentiation. In addition to myoblast fusion [26], IL-4 treatment also promoted myoblast differentiation in C2C12 cells (Figure 1A–C). In the IL-4-treated C2C12 cells, the expression of myogenic regulatory factors, MyoD and myogenin, and myomerger significantly increased, whereas the expression of myomaker was not affected (Figure 1F). Consistently, when we knocked down IL-4Rα expression in C2C12 cells, myoblast differentiation and fusion were suppressed (Figure 2) and the mRNA and the protein levels of myogenic regulatory factors and myomerger were significantly attenuated, whereas the expression of myomaker remained unchanged (Figure 3). These results suggest that the stimulation of the IL-4/IL-4Rα axis promotes myoblast fusion and differentiation, especially through the enhancement of MyoD, myogenin, and myomerger expression.

For the successful progression of the myogenic differentiation program, myoblasts must transition from the proliferation phase to the differentiation phase [31], during which MyoD and myogenin play essential roles [32]. Myogenin exerts its anti-proliferative effect by activating the genes that shut down the cell proliferation machinery [29]. The mRNA and protein levels of MyoD and myogenin, and the percentage of myogenin-positive C2C12 cells, were significantly higher in the IL-4 group than in the IL-4RαKD group (Figure 3). Moreover, we obtained the same results even when the cells were cultured in GM, a cell-proliferating condition (Figure 4). We also demonstrated that IL-4 treatment suppressed C2C12 cell proliferation (Figure 1E). All these results indicate that IL-4 signaling increases the expression of MyoD and myogenin and facilitates the transition of C2C12 cells from a proliferation phase to a differentiation phase.

MyoD and myogenin are transcription factors that promote the expression of myomaker and myomerger [19,20,21], which are essential membrane proteins for myoblast fusion [11,12,13,14]. In our study, the expression of myomerger increased by IL-4 treatment, whereas, unexpectedly, the expression of myomaker was not affected (Figure 1F). This apparent contradiction may be due that IL-4 signaling activated other regulatory mechanisms of myomaker transcription that may compete with or override the effects of MyoD and myogenin. The molecular links between IL-4 signaling and the regulation of myomaker and myomerger expression should be addressed in future studies.

For myoblast fusion, myomaker is required on both fusing myoblasts, but myomerger is required in only one cell of the fusing pair [15]. Thus, impaired myoblast fusion in the IL-4RαKD C2C12 cells should be caused by a reduction in myomerger expression. Contrary to our expectation, cell mixing assays and myomerger rescue experiments showed that the supplementation of myomerger was not sufficient to fully restore the fusion capacity of IL-4RαKD C2C12 cells (Figure 5). It is known that during myoblast fusion, many cellular events—including migration, adhesion, and cell–cell recognition—occur prior to myoblast membrane fusion wherein myomerger works [33]. Therefore, in addition to myomerger expression, the IL-4/IL-4Rα axis should also regulate the molecules involved in the series of cellular events during myoblast fusion.

We showed that IL-4 treatment increased NFATc2 expression in an IL-4Rα-dependent manner and that NFATc2 knockdown abrogated the effects of activation of the IL-4/IL-4Rα signaling (Figure 6). On the other hand, it has been reported that NFATc2 enhances IL-4 expression and promotes the second phase of myoblast fusion that occurs between nascent myotubes and myoblasts [24]. Thus, there may be positive feedback between IL-4 and NFATc2 expression that strongly promotes myoblast fusion and differentiation. Although our results suggest that NFATc2 expression is enhanced by IL-4Rα activation, it is still unclear whether NFATc2 directly regulates the expression of myogenic regulatory factors and myomerger. Mechanistic roles of NFATc2 should be addressed in future studies. 

## 5. Conclusions

Our results provide the first evidence that IL-4 signaling promotes myoblast fusion and differentiation by increasing the expression of MyoD, myogenin, and myomerger. Since impairment of myoblast fusion causes several muscle disorders, the IL-4/IL-4Rα axis can be a potential therapeutic target for the treatment of muscular pathologies. 

## Figures and Tables

**Figure 1 cells-12-01284-f001:**
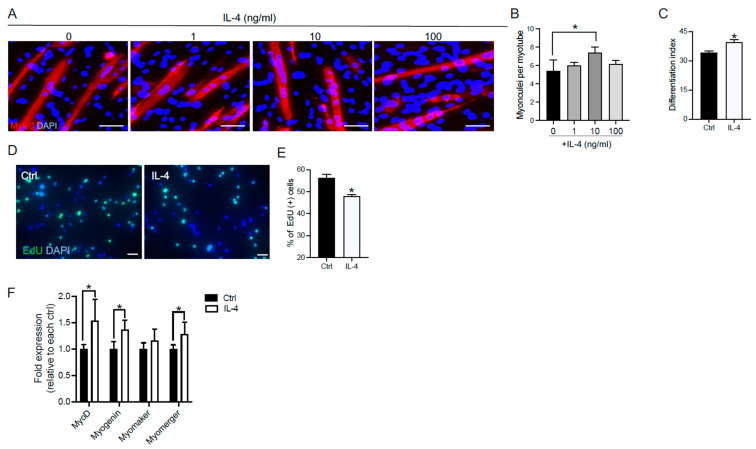
IL-4 treatment promoted myoblast differentiation and increased the expression of myogenic regulatory factors and myomerger in C2C12 cells. (**A**) Representative immunofluorescence images of MyHC (red) in the differentiated C2C12 cells 72 h after IL-4 application. The nuclei of C2C12 cells were counterstained with DAPI (blue). Scale bars: 50 μm. (**B**,**C**) The number of myonuclei per myotube and the differentiation index of IL-4-treated C2C12 cells. *N* = 3 per group. (**D**) Representative images of EdU (green) and counterstained nuclei (Hoechst 33342; blue) after 24 h IL-4 treatment in GM. (**E**) The number of EdU-positive cells significantly decreased in the IL-4-treated C2C12 cells. *N* = 4 per group. (**F**) Fold changes in the mRNA levels of MyoD, myogenin, myomaker, and myomerger in the IL-4-treated C2C12 myoblasts. Myoblasts were maintained in DM with 10 ng/mL IL-4. After 72 h incubation, the cells were harvested for qRT-PCR. *N* = 5 per group. Data are presented as mean ± SD. * *p* < 0.05.

**Figure 2 cells-12-01284-f002:**
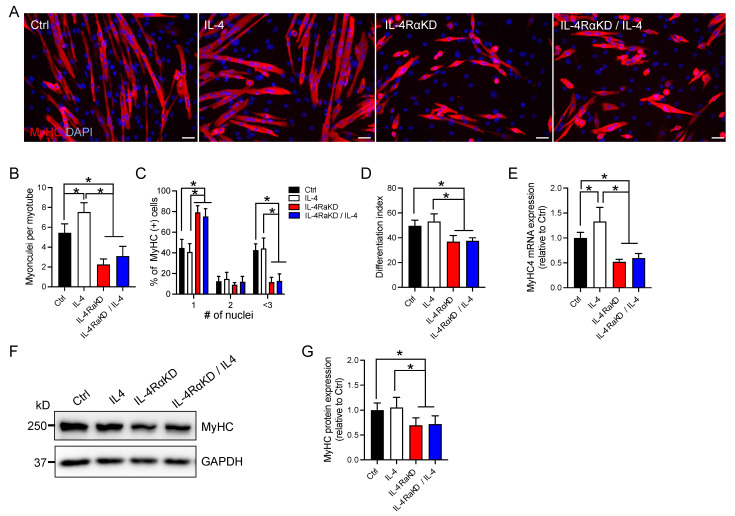
IL-4Rα knockdown impaired myoblast fusion and differentiation in C2C12 cells. (**A**) Representative immunofluorescence images of MyHC (red) in the IL-4RαKD C2C12 myoblasts. Cell nuclei were counterstained with DAPI (blue). Myotube formation was inhibited in the IL-4RαKD C2C12 cells. In this experiment, C2C12 cells were transfected with control siRNA (Ctrl) or IL-4Rα siRNA (IL-4 RαKD) and grown in GM for 24 h. The medium was replaced by DM with or without recombinant IL-4 (10 ng/mL). After 72 h incubation, cells were fixed and stained. Scale bars: 50 μm. (**B**–**D**). In (**B**–**D**), the number of myonuclei per myotube; the percentage of myosin-positive cells containing 1, 2, or ≥3 nuclei; and the differentiation index were shown, respectively. These parameters of myoblast differentiation significantly decreased by IL-4Rα knockdown. *N* = 5 per group. (**E**–**G**) Reduction of IL-4/IL-4Rα signaling by IL-4Rα knockdown significantly suppressed the increased expression of MyHC. Cells were treated as described in (**A**). (**E**,**G**) show fold changes in mRNA and protein levels, respectively. *N* = 6 per group. Representative western blot analysis is shown in (**F**). *N* = 6 per group. Data represent mean ± SD. * *p* < 0.05.

**Figure 3 cells-12-01284-f003:**
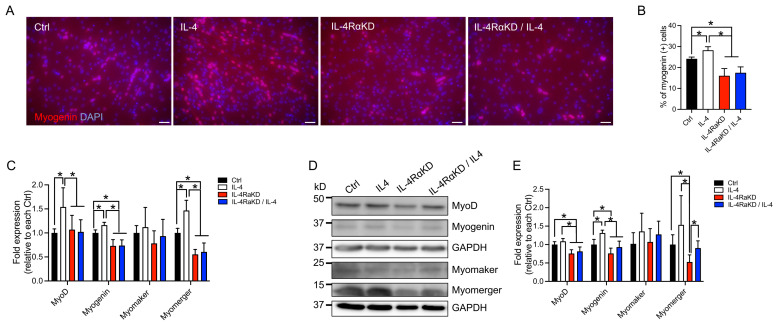
Reduction of IL-4/IL-4Rα signaling suppressed the increased expression of MyoD, myogenin, and myomerger, but did not affect myomaker expression. (**A**) Representative immunofluorescence images of myogenin (red) in the IL-4RαKD C2C12 myoblasts. Cell nuclei were counterstained with DAPI (blue). C2C12 cells transfected with control siRNA (Ctrl) or IL-4Rα siRNA (IL-4RαKD) were grown in GM for 24 h and maintained in DM with or without recombinant IL-4 (10 ng/mL) for 72 h. Scale bars: 50 μm. (**B**) The percentage of myogenin-positive cells was significantly suppressed by IL-4Rα knockdown. *N* = 5 per group. (**C**–**E**) Reduction of IL-4/IL-4Rα signaling by IL-4Rα knockdown significantly suppressed the increased expression of MyoD, myogenin, and myomerger, whereas the expression of myomaker was not affected. Cells were treated as described in (**A**). (**C**,**E**) show fold changes in mRNA and protein levels, respectively. *N* = 6 per group. Representative western blot analysis is shown in (**D**). Data represent mean ± SD. * *p* < 0.05.

**Figure 4 cells-12-01284-f004:**
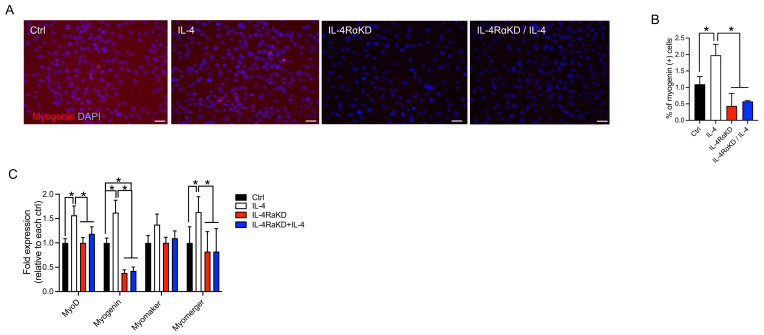
Stimulation of IL-4/IL-4Rα signaling increased the expression of myogenic regulatory factors and myomerger, but not myomaker, even in GM. (**A**) Representative immunofluorescence images of myogenin (red) in the IL-4RαKD C2C12 myoblasts. The nuclei were counterstained with DAPI (blue). C2C12 cells were treated as in Figure 3 except that cells were maintained in GM, instead of DM, for 24 h. Scale bars: 50 μm. (**B**) The percentage of myogenin-positive cells significantly increased by IL-4 treatment, which was suppressed by IL-4Rα knockdown. *N* = 4 per group. (**C**) Fold change in the mRNA levels of MyoD, myogenin, myomerger, and myomaker. *N* = 6 per group. Data represent as mean ± SD. * *p* < 0.05.

**Figure 5 cells-12-01284-f005:**
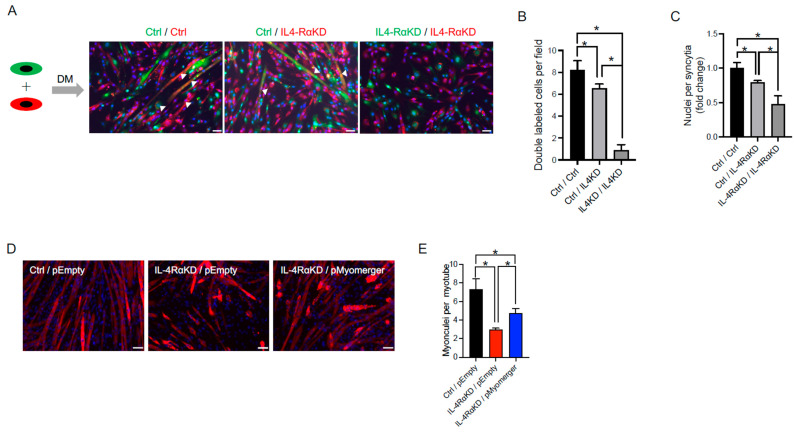
Supplementation of myomerger partially restored the impaired myoblast fusion capacity in IL-4RαKD C2C12 cells. (**A**–**C**) In a cell mixing assay, C2C12 cells were transfected with control (Ctrl) or IL-4Rα siRNA (IL-4RαKD) and labeled with green or red fluorescent lipid. Equal numbers of cells with each color were mixed and differentiated in DM for 72 h. (**A**) Representative images of cell mixing experiments. Arrowheads indicate double-labeled syncytia. Scale bars: 50 μm. (**B**,**C**) The number of double-labeled cells and the number of nuclei per syncytia were significantly increased in the Ctrl/IL-4RαKD group when compared to the IL-4RαKD/IL-4RαKD group. However, it did not reach the level of the Ctrl/Ctrl group. *N* = 5 per group. (**D**,**E**) One day after siRNA transfection, a myomerger expression vector (pMyomerger) or a control vector (pEmpty) was transfected. After 72 h of differentiation in DM, myoblast cells were immunostained or subjected to Western blotting analysis. (**D**) Representative immunofluorescence images of MyHC (red). Cell nuclei were counterstained with DAPI (blue). Scale bars: 50 μm. (**E**) The number of myonuclei per myotube in the IL-4RαKD/pMyomerger group was significantly greater than that in the IL-4RαKD/pEmpty group but still significantly less than that in the Ctrl/pEmpty group. *N* = 4 per group. Data are presented as mean ± SD. * *p* < 0.05.

**Figure 6 cells-12-01284-f006:**
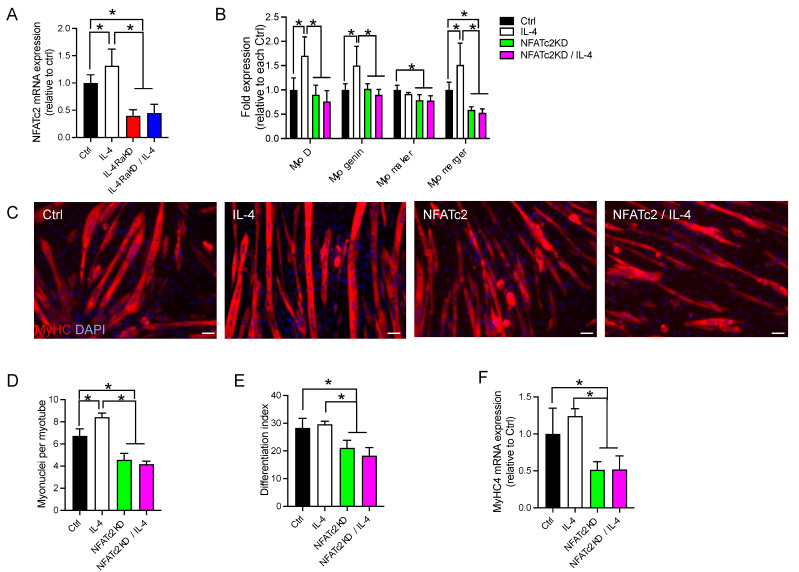
Knockdown of NFATc2 abolished the effects of activation of the IL-4/IL-4Rα signaling pathway in C2C12 cells. (**A**) IL-4 treatment promoted the expression of NFATc2 mRNA. Fold changes in the mRNA levels of NFATc2 are shown. (**B**) Knockdown of NFATc2 significantly suppressed the increased expression of MyoD, myogenin, myomaker, and myomerger. The fold changes of their mRNA levels are shown. For the experiments in (**A**,**B**), C2C12 cells were transfected with control siRNA (Ctrl) and IL-4Rα siRNA (IL-4RαΚD) or NFATc2 siRNA (NFATc2KD) and were grown in GM for 24 h. The medium was replaced by DM with or without recombinant IL-4 (10 ng/mL), and the cells were maintained for 72 h. Total RNA was extracted and subjected to qRT-PCR analysis. *N* = 3–4 per group. (**C**) Representative immunofluorescence images of MyHC (red) in the NFATc2KD C2C12 myoblasts. Cell nuclei were counterstained with DAPI (blue). Myotube formation was inhibited in the NFATc2KD C2C12 cells. In this experiment, cells were treated as described above. Scale bars: 50 μm. (**D**–**F**) The number of myonuclei per myotube, the differentiation index, and fold changes of MyHC4 mRNA are shown in (**D**–**F**), respectively. These three parameters of myoblast fusion and differentiation were significantly decreased by NFATc2 knockdown. *N* = 3–4 per group. Data represent mean ± SD. * *p* < 0.05.

## Data Availability

The data presented in this study are available upon request from the corresponding author.

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
