# Peer review of "IL-4 Signaling Promotes Myoblast Differentiation and Fusion by Enhancing the Expression of MyoD, Myogenin, and Myomerger"

_cells, 2023, doi:10.3390/cells12091284_

Round 1

Reviewer 1 Report

Summary:

Myoblast fusion is essential for skeletal muscle development, growth, and regeneration. Impairment of myoblast fusion causes several muscle disorders. Previously, Mitsutoshi Kurosaka, et al. have shown that IL-4 treatment promoted myoblast fusion in vitro, but the underlying mechanisms remain unclear. In this study, Mitsutoshi Kurosaka, et al used C2C12 myoblast cells to unravel that IL-4 treatment induces the expression of the key proteins associated with myoblast fusion and differentiation, such as myoblast determination protein 1 (MyoD) and myogenin, as well as their downstream target myomerger, an essential membrane protein for myoblast fusion. Furthermore, they provided evidence that this effect of IL-4 treatment was mediated by IL-4Rα. These findings suggest that the IL-4/IL-4Rα axis can be a potential therapeutic target for the treatment of muscular pathologies.

General comments:

This manuscript is well-structured. The introduction provides sufficient background. Materials & Methods are adequately described. Experimental design is appropriate to pursue the aims of this research and the methodology is sound. The results are clearly presented and precisely interpreted.

Author Response

Reviewer comment: This manuscript is well-structured. The introduction provides sufficient background. Materials & Methods are adequately described. Experimental design is appropriate to pursue the aims of this research and the methodology is sound. The results are clearly presented and precisely interpreted.

Our response: We are grateful for the comments. Thank you very much for finding our study interesting.

Reviewer 2 Report

This manuscript by Kurosaka et al. (entitled: IL-4 signaling promotes myoblast differentiation and fusion by 2 enhancing the expression of MyoD, myogenin, and myomerger) investigates the role of IL-4 in myoblast fusion.
Interestingly, Horsley et al. (CELL 2003) reported already "transcription factor NFATc2 controls myoblast fusion at a specific stage of myogenesis after the initial formation of a myotube and is necessary for further cell growth. By examining genes regulated by NFATc2 in muscle, this study identifies the cytokine IL-4 as a molecular signal that controls myoblast fusion with myotubes. Muscle cells lacking IL-4 or the IL-4α receptor subunit form normally but are reduced in size and myonuclear number. IL-4 is expressed by a subset of muscle cells in fusing muscle cultures and requires the IL-4α receptor subunit on myoblasts to promote fusion and growth. "

First:
In this context, it would be interesting to know whether it is transcription factor NFATc2 involved in transcriptional regulation of the expression of MyoD, myogenin, and myomerger. Chromatin Immun Precipitations should be done to find out whether NFACTc2 binds the promoters of MyoD, myogenin, and myomerger.

Second:
Additional data should confirm myotube formation by looking for Myosin heavy chain expression as a differentiation marker by qPCR and protein expression

Minor:

row
161   differentiationby

Author Response

POINT-BY-POINT RESPONSE TO THE REVIEWERS’ COMMENTS

We would like to thank the reviewers for their time and effort in reviewing our manuscript. The reviewers’ comments were highly constructive and significantly improved the quality of our study. All major changes done in response to the reviewers’ suggestions have been highlighted in this revised submission. The following are point-by-point responses to the reviewers’ comments.

***********************************************************

REVIEWER #1

Reviewer comment: This manuscript is well-structured. The introduction provides sufficient background. Materials & Methods are adequately described. Experimental design is appropriate to pursue the aims of this research and the methodology is sound. The results are clearly presented and precisely interpreted.

Our response: We are grateful for the comments. Thank you very much for finding our study interesting.

REVIEWER #2

Reviewer comment: In this context, it would be interesting to know whether it is transcription factor NFATc2 involved in transcriptional regulation of the expression of MyoD, myogenin, and myomerger. Chromatin Immun Precipitations should be done to find out whether NFATc2 binds the promoters of MyoD, myogenin, and myomerger.
Our response: Thank you very much for this important comment. NFAT proteins regulate the expression of many cytokines including IL-4 (Rao et al., 1997). Horsley et al. (2003) demonstrated that NFATc2 stimulates the production of IL-4, which mediates myoblast fusion. Thus, NFATc2 should be upstream of IL-4 signaling pathway. In the previous manuscript, we did not clearly describe about this point, so we may have misled the reviewers. We have made the appropriate correction in the revised manuscript (please see lines 59-60 in Introduction section).

Since the purpose of our study is to determine the downstream targets of IL-4 signaling for myoblast differentiation and fusion, we did not focus on NFATc2. However, the possibility remains that the IL-4 signaling pathway may stimulate NFATc2 and form a positive feedback loop that promotes the expression of myogenic regulatory factors and myomerger. We understand that the Ch-IP assay is the most direct and solid method to demonstrate that NFATc2 regulates the expression of MyoD, myogenin and myomerger. However, Ch-IP was not established in our laboratory and we had limited time for a major revision. Therefore, we tried to answer the reviewer's comment with the NFATc2 knockdown experiment. (We have placed related data as Fig. 6, Supplementary Fig. 3, and Supplementary Table 1. Please see also Material and Methods (lines 90-91, 138), Results (lines 266-284), and Discussion (lines 327-336) of the revised manuscript.)

We first investigated whether IL-4 treatment increases NFATc2 mRNA expression in C2C12 cells. We showed that the expression of NFATc2 mRNA was significantly higher in the Ctrl and IL-4 groups than in the IL-4RαKD and IL-4RαKD/IL-4 groups. Next, we examined the effects of NFATc2 knockdown. The mRNA expression of MyoD, myogenin, and myomerger was significantly lower in the NFATc2KD and NFATc2KD + IL-4 groups compared to the IL-4 group, whereas myomaker mRNA expression was unaffected by NFATc2 knockdown. Although we could not confirm whether NFATc2 can bind the promoter regions of MyoD, myogenin and myomerger, these results suggest that NFATc2 may directly or indirectly regulate their expression.

Reviewer comment: Additional data should confirm myotube formation by looking for Myosin heavy chain expression as a differentiation marker by qPCR and protein expression.

Our response: We appreciate the comments. The revised manuscript now includes data from Western blotting and qRT-PCR of MyHC. The results clearly show that cells treated with IL-4 increase both the mRNA and protein expression levels of MyHC compared with IL-4RαKD cells. We placed related data as Fig. 2 and Supplementary Table 1. Please see also Material and Methods (line 135 and), Results (lines 199-200) and Figure legends of Fig. 6 in the revised manuscript.

Reviewer comment: row 161  differentiationby.

Our response: We have corrected a typo. Please see line 166.

References

Rao, A., Luo, C., and Hogan, P.G. (1997). Transcription factors of the NFAT family: regulation and function. Annu. Rev. Immunol. 15, 707–747.

Horsley V., Jansen M. K., Mills T. S., Pavlath K. G. (2003). IL-4 acts as a myoblast recruitment factor during mammalian muscle growth. Cell, 113, 483-494. 2003.

Reviewer 3 Report

The authors investigated the roles of IL-4 signaling on C2C12 myoblast differentiation and fusion and founded that IL-4 promoted both of differentiation and fusion through IL-4 receptor alpha (IL-4Rα). Moreover, the authors observed that IL-4/IL-4Rα signaling regulated the expression of MyoD and myogenin for differentiation, and myomerger for fusion. The overexpression of myomerger partially rescued the impaired fusion in the IL-4Rα knockdown cells. Therefore, the authors concluded that IL-4/IL-4Rα signaling promotes C2C12 differentiation and partially regulates C2C12 fusion. The manuscript newly suggests that the molecular mechanism of IL-4 on skeletal muscle development. In addition, this work is very carefully carried out and, overall, the data are convincing.

Comments:

I could not confirm the method for “differentiation index (page4, line 162)”. The authors should the method.

Author Response

Reviewer comment: I could not confirm the method for “differentiation index (page4, line 162)”. The authors should the method.

Our response: We are grateful for the comments. Thank you very much for finding our study interesting. As for “differentiation index”, please see lines 128-129 in Material and Methods of the revised manuscript.

Round 2

Reviewer 2 Report

The manuscript has been revised quite well and is now ready for acception.